# The Ethics of Ancient Lactation and the Cult of the Perfect Breastfeeding Mother

**DOI:** 10.3390/healthcare11222941

**Published:** 2023-11-10

**Authors:** Elisa Groff, Florian Steger

**Affiliations:** Institute of the History, Philosophy and Ethics of Medicine, Ulm University, 89081 Ulm, Germany; florian.steger@uni-ulm.de

**Keywords:** ethics of lactation, ancient medicine, ancient philosophy, lactation disorders, exclusive breastfeeding, mental health nursing, postpartum

## Abstract

Breastfeeding is a key issue found in ancient sources that resonates with public debates today, affecting women in different parts of the world and of all social classes. The aim of this research was to identify breastfeeding narratives in ancient medical and philosophical texts from the 1st to the 6th century CE that address ethical issues in the medical management and social perception of new mothers. We examined 15 literary sources and one funerary inscription on lactation and critically evaluated the ancient idea of the perfect breastfeeding mother versus the non-breastfeeding mother. We then discussed our historical data in terms of objectivity and significance in relation to contemporary attitudes towards motherhood and lactation, e.g., (1) the cult of the perfect, breastfeeding mother in contemporary lactation education and (2) the onset of conditions which may affect normal breastfeeding, such as dysphoric milk ejection reflex (D-MER), breastfeeding aversion response (BAR) or post-partum depression. The analysis of the results showed that in both ancient and contemporary postnatal health care: (1) good mothering is associated with breastfeeding and (2) alternative feeding methods are acknowledged, but never as the best, natural option. Finally, our analysis shows that public health policies on breastfeeding and mothers’ own knowledge of their bodies are contested between nursing theories, social expectations and economic factors.

## 1. Introduction

Breastfeeding is a key issue found in ancient sources that resonates with public debates about babies’ right to mother’s milk, maternal ideology and mothers’ mental health that affect women today in different parts of the world and from all walks of life [1,2,3,4,5,6]. This article discusses the idea of the perfect breastfeeding mother in ancient and contemporary health discourses and parenting models by reflecting upon breastfeeding as a socially determined requisite for good mothering. It applies a comparative approach by looking at ancient philosophical and medical conceptualisation of breastfeeding and high-quality human milk in relation to contemporary attitudes to lactation within a socioecological framework [7,8]. The WHO and UNICEF recommend exclusive breastfeeding for the infant’s first six months, and continued breastfeeding with the addition of nutritious complementary solid food from seven months up to two years of age [9] (pp. 1–4), [10,11]. Reasons given in favour of human milk include short- and long-term health benefits, as well as positive psychological effects [5] (pp. 147–150), [9] (pp. 194–213), [12,13]. The benefits for the mother and for the baby are listed in Table 1 below.

Breastfeeding is often portrayed as a virtue in contemporary public health discourses, reflecting ideals of good mothering as well as women’s internalised social pressures [5] (pp. 101–116), [19,20,21], [22] (pp. 53–54). In recent years, scholarly articles have been published extensively on breastfeeding and motherhood in ancient religious and archaeological contexts [23,24,25], on lactation in Renaissance art and medicine [26] and also on the figure and skills of the most suitable wet nurse in antiquity [27,28]. Contemporary theories of lactation as both a biological process and a cross-cultural phaenomenon have also been explored in extensive publications from medical [9], nursing [29], psychological [9] (pp. 194–213) and socio-anthropological [5,30] perspectives. However, puerperal and domestic scenarios in which mothers experience discomfort or physical difficulties with breastfeeding do not often make the headlines in either ancient or contemporary texts [3,30,31,32,33,34,35,36].

Therefore, by critically analysing ancient medical and philosophical narratives on breastfeeding from the 1st to the 6th century CE, this article poses a primary question: were non-breastfeeding mothers considered “bad mothers” in antiquity? Accordingly, it examines (1) what happened when a mother could not or chose not to breastfeed her baby in antiquity; and (2) whether ancient theory, practice and culture diverged on the ethics of lactation.

Finally, a limitation of this paper lies in its focused approach. However, a holistic overview of the phenomenon “breastfeeding” from antiquity to the present day is beyond the scope of the present paper. Our research aims to provide a focused perspective on the ethical issues of breastfeeding in terms of: (1) timeframe: the 1st to 6th century CE; (2) objective: difficult and discriminatory experiences of breastfeeding; (3) subjective perception: looking at the experience of breastfeeding from the mother’s point of view; and (4) objective perception: reflecting the clash between the mother’s self-perception of her body and experience of breastfeeding and that of the society. By investigating the relationship between the new mother’s agency and her environment in a historical perspective, the article shows that historically: (1) ethical issues revolving around maternal and new-born health belong to the cultural contexts of health and wellbeing, and (2) new mothers are often caught up between the social acceptance of breastfeeding as a virtue, medical normativity, their own postpartum body and mental health [5,30,35], [37] (p. 243), [14] (p. 120), [38] (p. 79), [39,40].

## 2. Materials and Methods

This article is an analysis of ethical issues related to breastfeeding in ancient health discourses. The aim of this research was to identify breastfeeding narratives in ancient medical and philosophical texts that address ethical issues in the medical management and social perception of new mothers. Therefore, we analysed 15 literary sources and 1 funerary inscription on lactation from the 1st to the 6th century CE. The search was carried out in the online databases of the *Library of Latin Texts* (LLT: http://clt.brepolis.net/llta/pages/Search.aspx, accessed on 17 July 2023) and the *Thesaurus Linguae* Graecae (TLG: http://stephanus.tlg.uci.edu/index.php, accessed on 17 July 2023). The following keywords were retrieved in the LLT through a lemmatic search: *mater*, *matris* (“mother”); *lac*, *lactis* (“milk”); *contra naturam* (“against nature”); *integra mater* (“good mother”). The search algorithm in the TLG included the following lemmata: γάλα, γάλακτος, τὸ (“milk”); βρέφος, βρέφους, τὸ (“newborn”); γυνή, γυναικός, ἡ (woman, mother); μαστός, μαστοῦ, ὁ (“breast”); ἀποστήμα ἐν μαστοῖς (“abscess in the breasts”); πρὸς τὰς σκληρυνομένας φλεγμονὰς μαστῶν (“about the indurated, inflamed breasts”); θηλή, θηλῆς, ἡ (“nipple, breast”); θηλάζω (“to provide milk, to suckle”); θήλασμα, θηλάσματος, τὸ (“breastfeeding”). We critically evaluated the different contexts presented in our sources, in which breastfeeding is (1) the ideal nutrition for the newborn infant (Soranus, *Gynecology,* 2.11.18; Pliny the Elder, *Natural History* 28.21.72–73); (2) rejected as a personal, family choice, for which wet nursing is given as an alternative method (Aulus Gellius, *Attic Nights* 12.1.4–6, 22–23; Pseudo-Plutarch, *The Education of Children* 3d5), but deplored as “unnatural”; or (3) hindered by an inflammation or infection of the breast (Aetius of Amida, *Tetrabiblos* 16.34–39). Finally, we discussed our historical data in terms of objectivity and significance in relation to contemporary attitudes towards motherhood and lactation [41]. We then examined contemporary conceptualisations of the puerperal female reproductive body [21] and reasons for not initiating breastfeeding—e.g., in the context of dysphoric milk ejection reflex [33], breastfeeding aversion response [31] or postpartum depression—in the following contemporary sources: (1) clinical and social studies, (2) medical protocols for breastfeeding, (3) guidelines for the clinical care of new mothers and (4) interviews [3,6,13], [42] (p. 143), [43,44]. An overview of Section 2 is provided in Table 2 below.

## 3. Results

Ancient medical and philosophical sources from the 1st to the 6th centuries CE vary in the degree to which breastfeeding and good mothering compute. As Table 3 shows, our primary sources deal with (1) the value of high-quality milk for the nourishment of the baby; (2) breastfeeding as a virtue and as a medium of relational bonding; (3) the malfunctioning of breastfeeding due to physical or psychological causes; and (4) the decision not to breastfeed.

### 3.1. The Nutritional Value of Human Milk: Texture and Health Benefits

The ancient Roman philosopher and author, Pliny the Elder, states in his *Natural History* that human milk is “the sweetest and most delicate of all” drinks (28.21) [47] (p. 72). Moreover, he adds, (1) “for all purposes, a woman’s milk is more efficacious if she has given birth to a boy”; (2) “the most efficacious” of all is the milk of a new mother, “who has borne twin boys and abstains herself from wine and more acrid foods” [47] (p. 72); (3) while “the milk of a woman […] who has borne a girl is excellent, but only for curing spots on the face” [47] (p. 73). According to the ancient Greek physician Soranus of Ephesus (2.11.17), the baby must be deprived of “all food up to as long as two days” after birth, as he or she needs to recover from the violent shock of birth [48]. This recommendation reveals the ignorance of ancient medicine about colostrum after birth, even though the Greek word “*púos*”, “first milk after birth”, is known in ancient Greek language and is mentioned, for example, in the ancient Athenian comedy *Peace* (1150) by the playwright Aristophanes, and as “the purest of milk” in his play *Wasps* (710). Depriving babies of colostrum was a massive nutritional disadvantage for women and babies in ancient times, one that dramatically reduced their chances of survival. This is all the more relevant given that infant mortality rates were extremely high in antiquity [36], [49] (pp. 107, 109).

Ancient medical recommendations may not be concerned with colostrum, but they are concerned with the properties and texture of human milk, which must be checked for: (1) taste: it must be consistent and not change over time; (2) colour: it must be white; (3) smell: it must be pleasant; (4) composition: it must be smooth and homogenous; (5) density and thickness: watery is not good (see Soranus 2.13.22: [48]; Aetius of Amida 4.5: [50]). Interestingly, the ancient physician Soranus, whose work addressed the Roman elite household, refers here to the milk of wet nurses, which, like the wet nurse herself, had to be put under scrutiny before being employed.

### 3.2. Breast Is Best

In Book 12 of his commonplace work *Attic Nights*, the first-century Roman author, Aulus Gellius, tells his readers about his friend, the sophistic philosopher Favorinus, and his view of the fundamental value of breastfeeding for the mother’s mental health, the mother–baby bond, and the baby’s growth and cognitive development. Aulus Gellius reports that he himself once witnessed Favorinus visiting a friend and disciple on the occasion of the birth of his son. During his visit, Favorinus asked about the condition of the new mother and whether she intended to breastfeed her baby herself. When he learnt from the mother of the puerpera that she would not, Favorinus expressed his deep concern as follows: “I beg you, woman, let the girl be wholly and entirely the mother of her own child. For what kind of unnatural (*contra naturam*), imperfect and half-motherhood is to bear a child and at once send it away from her? To have nourished in her womb with her own blood something which she could not see, and not feed with her own milk what she sees, now alive, now human, now calling for a mother’s care?” (12.5–7: [51]). Furthermore, Favorinus adds: “Is the blood which is now in the breasts not the same that it was in the womb, merely because it has become white from abundant air and warmth? […] Therefore, it is believed not without reason that, just as the power and nature of the seed are able to form likeness of body and mind, so the qualities and properties of the milk have the same effect” (12.12–15 [51]).

Favorinus’ advocacy of exclusive breastfeeding as essential for both maternal bonding and the physical–cognitive development of the baby extends so far as to associate wet nursing with bereavement: “What the mischief, then, is the reason for corrupting the nobility of the body and mind of a newly born human being, formed from gifted seeds, by the alien and degenerate nourishment of another’s milk? Especially if she whom you employ to furnish the milk is either a slave or of servile origin […]” (12.17: [51]); “and there is no doubt that in forming character the disposition of the nurse and the quality of the milk play a great part […]” (12.20: [51]); furthermore, “when a child is given over to someone else and removed from the mother’s sight, the mother begins to gradually lose some of the strength of her maternal instinct […]. There is little difference between a woman who uses a wet nurse and a mother whose child has died” (12.22: [51]).

Breastfeeding as a woman’s virtue is also immortalised in a funerary inscription (CIL 6.19128) from the 2nd–3rd century CE dedicated to Gratia Alexandria by her husband, which reads: “TO GRATIA ALEXANDRIA, a woman of exemplary virtue and chastity, who nursed her children with her own milk […]”. Another philosophical author with whom Favorinus lived on close terms, Plutarch, reflects the same philosophical opinion that breastfeeding promotes bonding between mother and baby [52] (p. 87), but he also propagates the view that sharing milk means sharing long-lasting love and affection between milk-siblings. Plutarch does so by referring to Cato’s wife as an example of good mothering: “his wife always breast fed her son with her own milk. Indeed, she would often nurse the children of her slave in order to produce in them a natural love for her son since they had in infancy shared the same milk” (Plutarch, *Vitae parallelae*, Cato 20).

### 3.3. Choosing Not to Breastfeed

As mentioned in Section 3.2 above, the decision not to breastfeed is considered as “unnatural”, *contra naturam*, by the ancient philosopher Favorinus, because he found that it went against the normal physiology of the postpartum mother’s body and negatively affected the baby’s growth and social–cognitive development. However, although ancient philosophy and medicine agreed on the value of breastfeeding as the normative method of feeding, ancient medicine accepted wet nursing as an alternative legitimate method of feeding for the upper-class new mother. For example, the ancient Greek physician Soranus of Ephesus wrote (2.11. 7: [48] (p. 90): “Other things being equal, it is better to feed the child with maternal milk: for this is more suited to it, and the mothers become more sympathetic towards the offspring, and it is more natural, to be fed from the mother after parturition just as before parturition”. However, Soranus also adds: “if anything prevents breastfeeding, one must choose the best wet nurse, lest the mother grow prematurely old, having spent herself through the daily suckling”. In summary, the ancient medical writer expresses here a general preference for the mother’s milk, but he seems to be aware of how tiring breastfeeding can be for the mother and how it can lead to premature ageing. Therefore, if breastfeeding causes an overwhelming tiredness to the new mother, the family that can afford the advice should hire a wet nurse to perform the task. In the same vein, the ancient author Pseudo-Plutarchus testifies to the varying degrees to which breastfeeding and mothering were connected in ancient philosophy and medicine. In fact, in his treatise *The Education of Children* (3d5), he writes: “So, as I have said, mothers must endeavour, if possible, to nurse their children themselves; but if they are unable to do this, either because of bodily weakness (for such a thing can happen) or because they are in haste to bear more children, yet foster-mothers and nursemaids are not to be selected at random, but as good ones as possible must be chosen; and, first of all, in character they must be Greek” [53]. In sum, ancient philosophy and medicine promote breastfeeding as a maternal virtue, but also provide evidence of the refusal of breastfeeding as a family or personal choice, for which wet nursing and milk sharing are offered as alternative legitimate methods. This shows that different models of infant feeding and parenting were known in antiquity and that motherhood was conceptualised as a biocultural phaenomenon.

### 3.4. The Physiology of Breastfeeding

#### 3.4.1. Malfunctioning Breastfeeding Due to Psychological Complications

In chapter 12 of book 16 about “the management of the pregnant woman”, the medical author Aetius, native of Amida, today Diyarbakir in Turkey, begins his discourse with the following general admonition to the future physician: “pregnant women who have recently conceived must be protected from fear, sadness and all other kinds of mental disturbances. The patient should also avoid overactivity […], heavy weights and acrid foods […]” [54] (p. 22).

#### 3.4.2. Malfunctioning Breastfeeding Due to Physical Complications

Later in his discussion, Aetius devotes five chapters to distressing conditions associated with breastfeeding due to physical complications, such as the engorgement of the breast (16.35 and 16.36), mastitis (16.37), blocked milk ducts (16.38) and breast abscess (16.39). Moreover, he looks at the quality of contaminated human milk and its negative effects on the health of the infant (16.33) and deals with the management and prevention of blocked milk ducts (16.34). The late antique physician warns that the accumulation of milk in the breast should be prevented because: (1) it can seriously endanger the health of the child, which is an issue he had already discussed in an earlier book on paediatrics within his medical encyclopaedia (4.5) [50], and (2) it causes painful inflammation of the breasts, which can lead to obstruction of the mammary ducts and even abscesses. Therefore, in order to prevent the onset of this condition: (1) a proper diet should be prescribed and (2) “it is advisable to have the milk gently pumped by old and experienced women” [55] (p. 51). Indeed, as Aetius states, with reference to the second-century Greek physician Soranus: “if women breastfeed and children suckle, it helps relieve the pain of the breasts immediately, as suckling causes more milk to flow into the breasts” [55] (pp. 51–52). If the problem persists, a number of herbal remedies, plasters and exotic amulets are discussed at length. If nothing helps, it is necessary to resort to surgery and cut out the abscess [55] (p. 57). Breast inflammation is also mentioned in the hagiographical text that recounts the martyrdom of two new mothers, Felicitas and Perpetua, in the third century AD. In addition to describing scenes of breastfeeding in prison and of Felicitas’ breast dripping milk as she entered the arena, the hagiographer notes that Perpetua’s “breasts did not become inflamed” when she stopped breastfeeding her baby, nor did she acquire “sore breast” (6.6) [56].

## 4. Discussion

Breastfeeding is accepted as normative in all societies [7]. There is, however, a wide range of breastfeeding experiences. Some work out well, while some do not, and others require alternative feeding methods or emergency options [31,57]. Problems can be triggered by a cascade of con-causes [9] (p. 209). As every perinatal experience is different, breastfeeding can be both empowering and traumatic.

### 4.1. The Value of Mother’s Milk: Health Benefits and Relational Bonding

Current evidence-based data confirms that colostrum is all a baby needs to survive the first few days of life, as it is packed with nutrients and vitamins to strengthen the child’s immune system [9] (pp. 95–100), while “breast milk contains all the nutrients that an infant needs for the first 6 months of life, including fat, carbohydrates, proteins, vitamins, minerals and water” [9] (pp. 92–145), [7,12,15,58,59]. Although the ancient gynaecologist Soranus did not discuss colostrum, he did pay close attention to the properties and texture of human milk in general, and of that of wet nurses in particular. Today, we know that the continuous milk production is controlled by hormones that regulate milk supply and demand [34]. Ancient doctors were unaware of the biochemical dichotomy between oxytocin and prolactin, and medical texts from antiquity inform us of the wronged understanding of the endocrinology of lactation in medical knowledge until the Renaissance. In fact, ancient Greek and Roman authors mistakenly believed that there was a vascular connection capable of transferring concocted menstrual blood from the uterus to the breasts, where it was thought to be transformed then into milk (Corpus Hippocraticum, *Epidemics* 2.17 = 5.118 Littré; Aristotle, *History of Animals* 3.21 = 523a.1-13; Galen, *Hygiene* 1.7.19 = 6.36 Kühn; Galen, *On the Usefulness of the Parts of the Body* 16.10 = 2.419).

Today there is a consensus amongst clinicians, lactation consultants and social support services that “on the healthy end of the spectrum, breastfeeding fosters maternal responsiveness and promotes healthy mother–baby attachment” [34] (p. 3). Although ancient philosophers, doctors and families were not familiar with the biochemistry of breastfeeding, hormones and blood circulation, they seem to have known about the causes and effects of these hormones in the bodies of mothers and babies, leading to a bond—as Favorinus’s plea for breastfeeding and his dramatic association of wet nursing with bereavement illustrate (see Section 3.2 above).

### 4.2. Choosing Not to Breastfeed

In contemporary medicine, when science validates a decision to breastfeed that comes naturally to the mother, it feels as the right course of action for the mother [5] (pp. 144–153, 162–181). However, a mother’s inability or refusal to breastfeed is also a scenario that should be contemplated in lactation education, although it may change over time. Furthermore, a lack of lactation should not be instrumentalised as “bad” against a version of “good motherhood” associated with exclusive breastfeeding, self-sacrifice and intensive parenting [22] (p. 59), [14] (p. 117), [5,60]. Some women do not breastfeed because of the sense of shame they associate with their naked, culturally eroticised breast [9] (pp. 209–210). Some women discontinue lactation earlier than they would like because they experienced discrimination when breastfeeding at work or in public [16], e.g., UK Equality Act 2010 (17.4). Furthermore, the internalised pressure experienced by new mothers is infused not only with the social value given to breastfeeding [3] (p. 167), [20], or with culturally determined stereotypes of “good mothering”, but also with tensions within different waves of feminism that dictate the profile of a truly good mother [5] (p. 221), [9] (p. 206), [19] (p. 308), [21]. Indeed, recent studies of contemporary breastfeeding models highlight that mothers often have to “engage in ideological work to cope with the disjuncture between the dominant cultural logic that breast is best” and their own “beliefs and experiences” of lactation [14] (pp. 117, 124–126), [39], [60] (pp. 149–154). In her research on attachment parenting and intensive motherhood in the UK and France, Charlotte Faircloth asked female respondents to her questionnaire to list some of the reasons they thought other women choose not to breastfeed their babies. The majority of the women replied that the reasons lie with these mothers being selfish, being non-committal and not prone to self-sacrifice. Two thirds mentioned a lack of clinical and psychological support in the perinatal period as a deciding factor [5] (pp. 218–219).

This shows that the choice to breastfeed or not and for how long is highly politicised in favour of different waves of feminism. This behaviour is counterproductive because: (1) it hinders a fruitful dialogue about breastfeeding policies, e.g., in lactation education or in the workplace [9] (pp. 765–945), [44], and (2) it obscures the real issue of the mother’s individual choice [3], [5] (pp. 218–219), [19] (pp. 308–309), [15,42].

### 4.3. Alternative Options to Mother’s Milk

If breastfeeding does not come naturally or if the mother chooses not to breastfeed, in fact, there are ways to support mother and baby with alternative options such as formula [9] (pp. 199–206), [14] (pp. 122–124), [45], milk banking [46], wet nursing or milk sharing [45,61,62,63]. These are biocultural phaenomena that can also be found in our ancient sources, which show that women’s sexuality and reproduction have always been issues belonging to the cultural context of health [60] (pp. 149–154), cf. [62] (p. 233). For example, the ancient narrative of Cato’s wife who used to also nurse “the children of her slave in order to produce in them a natural love for her son” resonates with contemporary experiences of human milk sharing. This in turn confirms that: (1) infant feeding is much more than the transfer of milk to sustain a baby’s life or to supplement another mother’s own milk [64], and (2) milk sharing facilitates interpersonal relationships between previously unknown people [61].

The WHO defines wet nursing as any infant-feeding where the baby is breastfed by someone in good health, who is not his or her mother [65]. Human milk sharing (HMS) is growing in popularity in the United States, for example, but little research has been conducted on the topic [64], e.g., La Leche League International. The practice is still non-normative and socially stigmatised, with both donors and recipients reporting challenges in: (1) accessing an HMS network and (2) obtaining medical and mental health support [2]. As a result, families relying on HMS remain often in the dark [64]. Another challenge that mothers who deviate from the “breast is best” ethos face today is the ostracism within their own peer group. In fact, mothers who practice exclusive breastfeeding as the only acceptable solution feel that they have to educate the mothers who do not share their idea of “intensive parenting” by matronising them in public [5] (pp. 31–33, 221), [14], [60] (pp. 149–156). For example, a 46-year-old mother of two, interviewed by Faircloth for her book *Militant Lactivism,* describes the peer discrimination she experienced when she decided to stop breastfeeding her four-year-old son: “I have lost a lot of friends and feel I am in a bubble world. I can only relate to others who have mothered the same” [5] (p. 221), cf. [60].

Thus, gynaecologists, obstetricians and lactation consultants have an ethical obligation to inform pregnant and postpartum women fully and comprehensively about the pros and cons of breastfeeding, as well as about alternative options if the mother cannot breastfeed [9] (pp. 897–900), [35]. For instance, bottle-feeding can provide a similarly powerful “emotional and behavioural experience” for both mother and baby [9] (pp. 199–206), [14] (pp. 122–126). Furthermore, peer-to-peer milk sharing has grown in popularity in recent years in the United States [63]. As part of the Anthropological Context of Milk Sharing Study (AnthroCOMS), Palmquist and Doehler interviewed 1116 mothers of at least 18 years of age, from 15 different countries, who reported that they were using or had used milk sharing because they had no access to banked donor milk [62] (p. 233). The AnthroCOMS study also highlighted that peer-to-peer milk sharing is still shrouded in moralised prejudices and misconceptions [66]. This demonstrates the need for guidelines to inform about shared donor milk, beyond stigma and misleading information [9] (pp. 765–945), [62] (p. 233), [67].

### 4.4. Lactation Disorders

A woman who chooses not to breastfeed because of physical or psychological distress deserves caring support and should not be labelled as “bad”. Indeed, for a mother who cannot breastfeed because of physical or physiological barriers, the question is not whether to maintain or abandon the idealistic choices she made about breastfeeding during pregnancy, but rather how not to betray (1) her baby’s “silent demand” in terms of initial trust and (2) her own—socially internalised—expectations of being a good mother without feeling guilty [9] (pp. 210–211), [22] (pp. 99, 107), [38] (p. 79), [68] (p. 117). This question is primarily ontological and belongs to the realm of ethical normativity. As recommended in our ancient medical sources, the skilled help of midwives, help providers or lactation consultants and the support of an intact family environment are considered fundamental in helping the mother struggling with “breast inflammation”, such as the delay in the onset of lactogenesis [9] (p. 79), dysphoric milk ejection reflex (D-MER) [32,34], breastfeeding aversion response (BAR) [31] or mastitis [9] (pp. 897–900), [69] (p. 843). Clinically, the term “breast inflammation” is not specific, as it does not distinguish between infectious and non-infectious processes. However, in postnatal care, it is used to refer to a spectrum of aetiologies, ranging from nipple infection to inflammatory cancer [70]. Lisa H. Amir reports for the Academy of Breastfeeding Medicine Protocol Committee that mastitis today affects between 3% and 20% of lactating women [69] (p. 840). This condition is associated with the early cessation of breastfeeding and the risk of abscess formation. As the primary cause is milk stasis, the prescribed treatment in contemporary postnatal care is frequent milk removal, accompanied and followed by a massage of the breast [69] (pp. 841, 843). It is worth noting that modern lactation theories support the ancient view that mother’s clogged milk is not bad for the baby, although some babies may not like the taste; whereas an early cessation of breastfeeding allows the germs left in the mother’s breast to spread in the milk [16]. As for breastfeeding aversion response (BAR), Mors et al. have recently shown that the causes of this phaenomenon are still unresearched. Little is known about the biochemistry, incidence and potential treatment of BAR. Therefore, there is a need for more targeted support for mothers experiencing BAR [31].

Similarly, mental health support during the antenatal and postnatal period should become central to breastfeeding support, and lactation consultants should be trained in recognising the symptoms of mood disorders [2,33] and the “potential co-morbidity of breastfeeding problems and postpartum depression” [3] (p. 167), [9] (pp. 208–209), [6,13,44], [62] (p. 234), [71] (pp. 919–928). Recent clinical publications recommend that women who experience antenatal discomfort and new mothers with breastfeeding difficulties should be monitored carefully and screened for depressive spectrum disorders [71,72]. Indeed, research confirms that the puerperium is a critical period for newborn and maternal health [72]. Clinical studies report that 30% to 80% of new mothers experience symptoms associated with a pregnancy-related mood disorders [73,74,75], 10% to 20% are diagnosed with postpartum depression and more than 50% remain undiagnosed [59,76,77]. Episodes of postpartum mood disorders can lead to the early cessation of breastfeeding, disrupt the mother–baby bond and affect the baby’s growth and brain development [71,72,74]. Maternal responses to perinatal depression and parental attachment vary according to: (1) the mother’s individual experience of breastfeeding, (2) her previous history of depression and general medical history, (3) the quality of the milk production, (4) the triggers of her own mood symptoms and (5) the domestic context in which mother and baby live. Some mothers with a diagnosis of depression or mood disorder experience relief from breastfeeding [59,78], while others find it a painful and traumatic process. Taking all this into account, the choice of treatment should be based on an individual risk–benefit analysis (Office of Disease Prevention and Health Promotion 2020): for example, for some mothers, the decision of taking an antidepressant proves to be the right one; for others, it would be arranging for another caregiver to bottle-feed the baby [71] (pp. 921, 922–925). Finally, current evidence highlights that it is crucial for both the mother’s recovery and maternal lactation status that the new mother is not merely labelled as “bad” or “depressed” [71] (p. 924).

### 4.5. Breastfeeding between Normativity and Practice

A woman who is experiencing difficulties in the puerperium deserves help tailored to her needs [9] (pp. 901–906), [32], especially as research has shown that perinatal depression does not necessarily affect breastfeeding intentions or initiation [33,57]. This implies that health providers should be responsive and adapt to the needs of mother and baby on an ad hoc basis, negotiating the best possible solution within the context of an ethics of care and equitable management of health resources [22] (p. 59), [3,7,79]. This approach paves the way for a model of healthcare that: (1) aims to strengthen people’s direct ability to act or their options to act and (2) is concerned with the quality of the relationship between those being supported and those providing care [6,80,81,82,83]. Indeed, in a caring relationship, we cannot escape the question of what is good, both for the patient and for the caregiver. However, what is good for the caregiver, based on what he or she holds to be good, should not override the needs and preferences of the patient [22] (p. 53), [79]. Therefore, health providers and lactation consultants should find ways to initiate an open conversation about the risks and benefits of all available alternatives to mother milk. In this way, maternal and newborn health can best be cared for and respected according to the health status and sociocultural circumstances of both mother and child [43] (p. 234), [45,84]. A mother’s “choice of feeding method does not make her a good or a bad mother” [9] (p. 210). From an ethical point of view, however, it is crucial that the choice is an informed one [9] (p. 211), [84].

While we should advocate breastfeeding, we should equally offer support to women who have chosen alternative feeding methods [14] (p. 124), [16]. If the mother is experiencing difficulties, she should be: (1) reminded that breastfeeding is a learning process for both the mother and her new baby and (2) advised to seek help from a midwife, a lactation consultant or a member of support services, e.g., www.breastfeeding.nhs.uk, accessed on 15 August 2023; www.laleche.org.uk, accessed on 15 August 2023, www.nationalbreastfeedinghelpline.org.uk, accessed on 15 August 2023. There is no shame in asking for help: we need to destigmatise the act of asking for help in order to facilitate a multiplicity of breastfeeding scenarios that fit more closely with the multiplicity of mothering experiences.

Finally, a lack of lactation, low milk production or a mother’s informed decision not to breastfeed often reflect the cultural context of health, sexuality and reproduction in which the mother and her baby are based. This also shows that new mothers are often caught between the normativity of breastfeeding as a virtue, cultural anxieties and social conceptualisations of their postpartum bodies that are divorced from their own bodily perceptions and mental health.

## 5. Conclusions

From a close analysis of the evidence, we found that: (1) in both ancient and contemporary postnatal health care, the “perfect new mother” is a “breastfeeding mother”; (2) medical and cultural contexts in which a mother cannot or chooses not to breastfeed are historically documented; (3) alternative feeding methods are contemplated, but never presented as the natural option, the best to which a baby is entitled. The main difference we observed is that, whereas ancient society invested in pro-breastfeeding social policies by always ensuring an alternative source of milk for the newborn, i.e., by implementing the availability of human resources in their community, such as wet nurses, current breastfeeding practices seem to be disconnected from the daily reality of mothers. This results in a lack of public policies to support the reconciliation of work and breastfeeding, to enable breastfeeding in public, to provide spaces for breast milk extraction and to normalise milk donation.

Our analysis is a call for a new paradigm to understand breastfeeding, both from a historical perspective and in the realm of contemporary postnatal healthcare. The essential point is that there is always another side to the best experience of breastfeeding. When we demonise the absence of breastfeeding per se, we insinuate that anyone who does not breastfeed is a “bad” mother [19,21]. When we demonise the choice of non-breastfeeding per se, we eliminate the practice of breastfeeding “as a right and a true choice for all women” [84]. Moreover, we remove the responsibility from those who are there to provide care, social policies and lactation education in our healthcare system. We should consider refocusing on ways to make breastfeeding a choice tailored to the needs of mother and baby, a choice that respects patient autonomy and ensures that mother and baby have a similar good bonding experience—in different cultural contexts of health and beyond stereotypes [9] (pp. 199–206), [35].

## Figures and Tables

**Table 1 healthcare-11-02941-t001:** Clinical reasons in favour of human milk.

Benefits for the Mother:	Benefits for the Baby:
Maternal bonding [14] (p. 119), [15] (p. 217), [16]. Economic savings [17]. Reducing the risk of breast and ovarian cancer [16]. Reducing the risk of cardiovascular disease [18]. Losing weight [14]. Helping the uterus to contract to its prenatal size [5] (p. 39), [14,16].	Maternal bonding. Reducing infections. Reducing the risk of sudden infant death syndrome [5] (p. 38), [9] (pp. 29–31). Reducing the incidence of obesity and diabetes later in life. Increasing the intelligence quotient score [14,16].

**Table 2 healthcare-11-02941-t002:** Ethical issues associated with breastfeeding in ancient and contemporary health discourses.

Ancient Material	Contemporary Issues	Ethical Reflections
The nutritional value of human milk: texture and health benefits (Pliny the Elder, *N.H.* 28.21; Soranus, 2.11.17, 2.13.22; Aetius, 4.5). Breast is best (Aulus Gellius, *Attic Nights* 12.5.7, 12–15, 12.20, 12.22; CIL 6.19128; Plutarch, *V.P.*, Cato 20. Choosing not to breastfeed (Soranus, 2.11.17; Pseudo-Plutarchus, *The Education of Children* 3d5). The physiology of breastfeeding: physiological and psychological causes of malfunctioning of the breast (Aetius of Amida, 16.33–39).	Breastfeeding and bonding. Breastfeeding and health benefits for the baby. Breastfeeding as a virtue. Physiological and psychological causes of malfunctioning breastfeeding [3]. Socially accepted alternatives to human milk: wet nursing, milk sharing, human milk banking or formula [9] (pp. 199–206), [14,45,46].	Mothers who cannot or who choose not to breastfeed are “bad”. Babies who are not breastfed are deprived of their basic right to nutrition [5] (p. 214). The first response to a mother’s nursing challenge should be to switch to an alternative source of milk, e.g., formula, wet nursing, milk sharing, human milk banking. The obstetrician/health provider should suggest that the mother stops breastfeeding if she is struggling to avoid feelings of guilt [38] (p. 79). Public and health professionals should put less pressure on exclusive breastfeeding.

**Table 3 healthcare-11-02941-t003:** Results: Ancient medical and philosophical sources dealing with aspects of breastfeeding (1st–6th century CE).

Subject	Medical Sources	Philosophical Sources
Section 3.1	Soranus, 2.11.17, 2.13.22; Aetius of Amida, 4.5	Pliny the Elder, *N.H.* 28.21.
Section 3.2	Soranus 2.11.18	Aulus Gellius, *Attic Nights* 12.5.7, 12–15, 12.20, 12.22; Plutarch, *V.P.*, Cato 20. Cf. CIL 6.19128.
Section 3.3	Soranus, 2.11.17	Pseudo-Plutarchus, *The Education of Children* 3d5.
Section 3.4.1	Aetius of Amida 16.12	
Section 3.4.2	Aetius of Amida, 16.33–39	

## Data Availability

Data are contained within the article.

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
