# Peer review of "The Ethics of Ancient Lactation and the Cult of the Perfect Breastfeeding Mother"

_healthcare, 2023, doi:10.3390/healthcare11222941_

Round 1

Reviewer 1 Report

Comments and Suggestions for Authors

The article offers a truly original approach by linking the sources of Antiquity with the conception of breastfeeding in the present time. This approach makes it possible to establish a genealogy of the elements that shape who is a good or a bad mother.

If the authors considered it appropriate, it would perhaps be worthwhile to delve into the main differences between ancient and present-day societies in the perception of breastfeeding. For example, it would be of interest to raise the double discourse existing in the contemporary context between the social value of breastfeeding, shared in Antiquity, and the reality of its current practice, which is greatly hindered by the denigration of breastfeeding in public, the lack of public spaces for breastfeeding and breast milk extraction, the lack of public policies that enable the work/breastfeeding compatibility.

From the formal point of view, I point out some  orthotypographic errors detected to facilitate the correction process:

Line 150:  double semicolon

Line 345: hat (instead of "that")

Line 592: Priary (instead of "Primary")

Author Response

Thank you very much for your kind comments and your precise orthographic observations, which we have corrected accordingly. We have added a new reflection in the “Conclusion” on the main difference we found between antiquity and the present day when it comes to investing in social policies and community resources to effectively promote breastfeeding (see lines 441-447).

“The main difference we observe is that whereas ancient society invested in pro-breastfeeding social policies by always ensuring an alternative source of milk for the newborn, i.e. by implementing the community human resources available, such as wetnurses, current breastfeeding practices seem to be disconnected from the daily reality of mothers. This results in a lack of public policies to support the reconciliation of work and breastfeeding, to enable breastfeeding in public, to provide spaces for breast milk extraction and to normalise milk donation”.

Reviewer 2 Report

Comments and Suggestions for Authors

The historical information is intriguing and useful as it underscores that breastfeeding attitudes/beliefs are culturally driven and should be individually supported to meet the needs of the breastfeeding dyad and family. One size does not fit all and history demonstrates that today's concerns are similar to yesterday's.

My major criticism is only having a few historical documents, but given that our breastfeeding issues are parallel, this article at least bridges that gap.

Comments on the Quality of English Language

No concerns with the English.

Author Response

Many thanks for your constructive comment. We have added a new paragraph in the “Introduction” to address the limitations and potential biases of the paper due to its focused historical approach, in order to better highlight the scope of the study (please see lines 63-70).

Reviewer 3 Report

Comments and Suggestions for Authors

Thank you very much for allowing me to review the article titled "The Ethics of Ancient Lactation and the Cult of the Perfect Breastfeeding Mother" (healthcare-2636704), which is submitted for the Special Issue "History, Philosophy, and Ethical Perspectives on Healthcare."

The aim of this research was to identify narratives about breastfeeding in ancient medical and philosophical texts that address ethical issues in the medical management and societal perception of new mothers.

This is an intriguing reflective article on a topic of significant interest due to its implications for both mothers and, more directly, infants. I have some suggestions to offer:

1. In the abstract, it is advisable to specify the time period under review and the sources of data used. This fundamental information will help connect previous research on this topic and future reviews.

2. The keywords used may pose difficulties in reproducing the review. Consider the possibility of using simpler words.

3. While the introduction is well-framed, the use of numerous bibliographic references together makes it challenging to assess their contributions. I believe the introduction should be rewritten, providing more detailed information instead of presenting it in an aggregated manner. The study's objective should be stated at the end of the introduction.

4. In the "Materials and Methods" section, the study's design should be defined, and the methodology should be clarified. It would be interesting to incorporate details about the databases used, the time period, and the methods employed for data extraction.

5. The results are presented in a reflective manner. Consider the possibility of including tables or graphs to enhance thematic comprehension.

6. The discussion, given the nature of this work, effectively forms part of the results. However, it should introduce future reflections, the current state of knowledge's weaknesses, and what we currently understand about breastfeeding.

7. The conclusions should be the contribution of the work, not merely a summary of it. Although the article presents some intriguing reflections that could transition to the discussion.

Author Response

Thank you very much for your helpful comments. We have added a new paragraph in the “Material Methods” to explain our modus operandi in terms of data retrieval and extraction (see lines 75-85). We also included a table in the “Results” section, as suggested, to summarise the results before explaning them. In general, the structure has been further clarified in the “Abstract” and in the “Introduction”, where the authors have foregrounded the temporal choices/ limitations of the study and their impact on the comparison (see added lines in the “Introduction”: 63-70). Finally, we have added a new reflection in the “Conclusion” highlighting the main difference between ancient and contemporary practices when it comes to investing in social policies to promote breastfeeding (see lines 441-447).

Reviewer 4 Report

Comments and Suggestions for Authors

The manuscript delves into the historical and cultural dimensions of breastfeeding, examining the societal expectations placed on mothers. While the subject matter is intriguing, the authors could significantly enhance the quality of their work by incorporating a more extensive range of current sources and references. This would not only bolster the reliability of their arguments but also provide a more comprehensive perspective on the ethics involved. Furthermore, it is essential for the authors to pay close attention to the limitations of their research. Historical data, potential biases, and the scope of the study should be transparently addressed to allow readers to assess the generalizability of the findings and their relevance to modern society. Recognizing these limitations would increase the credibility of the manuscript and provide a more holistic understanding of the topic.

Author Response

Thank you very much for your constructive and helpful comments.  

We are aware of the limitations of the paper’s historical approach. A holistic overview of the phenomenon “breastfeeding” from antiquity to the present day was beyond the scope of this paper. This research aims to provide a focused perspective on the ethical issues of breastfeeding in terms of (1) timeframe: 1st to 6th century CE, (2) objective: difficult and discriminatory experiences of breastfeeding, (3) subjective perception: looking at the experience of breastfeeding from the mother’s point of view, (4) objective perception: reflecting the clash between the mother’s self-perception of her body and experience of breastfeeding and that of the society. The points 1-4 were then discussed in relation to similar scenarios in current public health policy and lactation education practices. The reader is referred to the literature in the main text for further discussion and insights, and contemporary resources have been provided to bolster the argument: see in particular “Discussion” and “Introduction”). This structure has been further clarified in the abstract and introduction where the authors have foregrounded the temporal choices/ limitations of the study and their impact on the comparison: see lines added in the Introduction: 63-70.

Round 2

Reviewer 3 Report

Comments and Suggestions for Authors

I've reviewed the article titled "The Ethics of Ancient Lactation and the Cult of the Perfect Breastfeeding Mother" (healthcare-2636704) again, as well as the authors' response to the comments made. I believe the authors have clarified the points raised in the review and provide a holistic view of breastfeeding.

Comments on the Quality of English Language

No comments